# Molecular Mechanisms and Mediators of Hepatotoxicity Resulting from an Excess of Lipids and Non-Alcoholic Fatty Liver Disease

Carmine Finelli

Department of Internal Medicine, ASL Napoli 3 Sud, Via Marconi, 66, Torre del Greco, 80100 Napoli, Italy; carminefinelli74@yahoo.it

**Abstract:** The paper reviews some of the mechanisms implicated in hepatotoxicity, which is induced by an excess of lipids. The paper spans a wide variety of topics: from the molecular mechanisms of excess lipids, to the therapy of hyperlipidemia, to the hepatotoxicity of lipid-lowering drugs. NAFLD is currently the leading cause of chronic liver disease in Western countries; the molecular mechanisms leading to NAFLD are only partially understood and there are no effective therapeutic interventions. The prevalence of liver disease is constantly increasing in industrialized countries due to a number of lifestyle variables, including excessive caloric intake, unbalanced diet, lack of physical activity, and abuse of hepatotoxic medicines. Considering the important functions of cell death and inflammation in the etiology of the majority, if not all, liver diseases, one efficient therapeutic treatment may include the administration of hepatoprotective and anti-inflammatory drugs, either alone or in combination. Clinical trials are currently being conducted in cohorts of patients with different liver diseases in order to explore this theory.

**Keywords:** mediators of hepatotoxicity; lipids; hepatic statosis; toll-like receptors; lysosomal permeabilization; prostaglandins cyclooxygenase 2; ceramides; apoptosis

## 1. Introduction

It has been shown that hepatic steatosis, formerly thought to be the start of non-alcoholic fatty liver disease (NAFLD), is actually a healthy increase in triglycerides, whereas free fatty acids are the toxic molecules that lead to steatohepatitis and fibrosis [1,2]. In a mouse model, genetic suppression of the enzyme diacylglycerol acyltransferase 2 (DGAT2) decreased hepatic steatosis while increasing fibrosis due to the toxic effects of FFAs [3]. Lipotoxicity mediated by FFAs (cellular toxicity caused by fat accumulation) in liver cells may originate from the disruption of triglyceride synthesis [4].

NAFLD is a group of diseases that include cirrhosis, steatosis, and steatohepatitis, which are characterized by fatty infiltration of the liver. NAFLD is the most common liver disease, with an estimated prevalence worldwide of around 25% [5]. The current epidemic of obesity and metabolic syndrome, which manifest in the liver as NAFLD, may be responsible for the sudden increase in the incidence of NAFLD [6]. The majority of patients with NAFLD have simple hepatic steatosis without steatohepatitis and fibrosis. In addition, in 2–3% of individuals, NAFLD can become non-alcoholic steatohepatitis (NASH), which can lead to progressive fibrosis, cirrhosis, and consequences such as hepatocellular carcinoma [7–9]. As much as 50% of patients with simple steatosis who later develop NASH may also develop severe fibrosis [10].

The parenchyma of the liver is made up of a variety of cell types, the bulk of which are hepatocytes (around 70% of the liver cell population) [11]. Parenchymal cells include hepatocytes and cholangiocytes, while nonparenchymal cells consist of Kupffer cells, stellate cells, and endothelial cells. Hepatic cells orchestrate the development of liver disorders. Hepatocytes, macrophages, and hepatic stellate cells interact in NAFLD and

its severe variant, non-alcoholic steatohepatitis (NASH), although the precise methods through which these cells are orchestrated are not fully understood [11,12].

Patients with NAFLD have higher levels of oleic acid, a monounsaturated fatty acid (MUFA), and palmitic acid, according to studies examining the makeup of hepatic and circulatory free fatty acids (a saturated fatty acid (SFA)) [13,14]. However, polyunsaturated fatty acids have not been demonstrated to be toxic to hepatocytes and may even be beneficial in NAFLD patients [15,16]. Experimental research on this subject in human B cells has examined the function of stearoyl-CoA desaturase-1 (SCD1), the enzyme that changes SFA into MUFA [17]. More MUFAs are produced because of the increased expression of SCD1, and these MUFAs are then incorporated into triglycerides to produce simple and well-tolerated hepatic steatosis [18]. However, the removal of SCD1 results in the accumulation of SFA, which in turn triggers hepatocyte death and steatohepatitis [19]. Therefore, the type of FFAs stored in hepatocytes is just as important for developing NAFLD as the amount of FFAs accumulated, if not more [20]. Apoptosis, a form of programmed cellular death, is considered a key mechanism in developing NAFLD [21,22]. The main pathogenic mechanism seen in the biopsy samples of NASH patients is apoptosis, and in the spectrum of NAFLD, the presence of apoptosis helps to identify NASH patients from those with simple steatosis [23]. Patients who have higher levels of apoptosis will have advanced fibrosis because the degree of apoptosis and inflammation are inversely correlated [24]. The correlation between the levels of circulating cytokeratin-18 fragments, which are indicators of apoptotic liver cells, and the degree of fibrosis, provides additional evidence that apoptosis plays a significant role in NASH [25]. Lipoapoptosis is the name given to apoptosis mediated by FFAs [26]. Activation of the apoptotic pathways may occur either by an extrinsic pathway mediated by cell surface receptors or by an intrinsically mediated pathway by intracellular organelles [27].

A fundamental component of NASH is the lipotoxicity of hepatocytes. Lipotoxicity is caused by the accumulation of lipid intermediates, which cause cell malfunction and cell death. In NASH, hepatocytes build up triglycerides and various lipid metabolites, including free cholesterol, ceramides, and free fatty acids (FFA) [28]. Hepatocytes store most fatty acids as triglycerides, and some data suggest that the esterification of fatty acids into neutral triglycerides provides a protection mechanism against lipotoxicity [4,29]. On the other hand, FFA causes liver damage and activates specific signaling stunts, resulting in hepatocytic apoptosis, which in this context is called lipoapoptosis [30]. FFA is thus regarded as a major mediator of the lipotoxicity of hepatocytes. In fact, in NASH, there is an increased hepatic inflow of FFA following increased lipolysis in the peripheral fat tissue due to insulin resistance [31]. The lipotoxicity of FFA in hepatocytes is partially mediated through their intracellular lysophosphatidyl choline metabolite, which has also been seen to increase in the liver of patients with NAFLD in proportion with the severity of the disease [29].

Antihyperlipidemic drugs frequently cause mixed hepatocellular or liver lesions, with rare cases of pure cholestatic image [32,33]. The cytochrome P450 system, bile acid transport protein dysfunction, immune-mediated inflammatory response to the drug or its metabolites, immune-mediated apoptosis by tumor necrosis factor, and oxidative stress as a result of intracellular damage are some of the various proposed mechanisms of hepatotoxicity that vary depending on the drug or drug class [34].

The paper reviews some of the mechanisms implicated in the hepatotoxicity induced by an excess of lipids. The paper spans a wide variety of topics: from the molecular mechanisms of excess lipids, to the therapy of hyperlipidemia, to the hepatotoxicity of lipid-lowering drugs. NAFLD is now the leading cause of chronic liver disease in Western countries; the molecular mechanisms leading to NAFLD are only partially understood and there are no effective therapeutic interventions.

## 2. Mediators of Hepatotoxicity from Excess of Lipids

### 2.1. Toll-like Receptors

Toll-like receptors (TLRs) are pattern recognition receptors that are able to detect molecular forms associated with pathogens, and, as a result, they can activate the immune system by means of pro-inflammatory signaling pathways [35]. The production of adipocytokines such as TNF-α and IL-6 is high when TLR4-mediated upregulation of NF-κB is activated by saturated fatty acids such as palmitic acid [36]. A reduction in the expression of TLR4 in mutating mice has been shown to be protective against the development of NASH [37]. Mice that received high-fat diet and dextran sulfate sodium (DSS) had high levels of bacterial lipopolysaccharides in the portal circulation, a higher expression of TLR4, and severe liver inflammation when compared with the controls [38]. TLR4 may be the key component of the gut microbiota−liver axis, which influences the development of NASH [39]. In fact, TLR4 stimulation can stimulate the production of ROS by hepatic macrophages and improve the expression of pro-interleukin-1, eventually promoting the development of NASH [40]. This demonstrates how the etiology of NASH is affected by the impact of gut microbiota dysbiosis-mediated LPS/TLR4 activation.

### 2.2. Death Receptors

Death receptors, which are receptor family tumor necrosis factors on the cellular surface, are essential in extrinsic apoptotic pathways [41]. The liver expresses several death receptors and their ligands, including tumor necrosis factor receptor 1 (TNF-R1), TNF related apoptosis-inducing ligand receptor 1 and 2 (TRAILR1 and TRAIL-R2), Fas ligand (FasL), TNF-α, and TRAIL [42]. The death ligands in the extrinsic pathway bind their receptors to create a death complex, which then activates caspase-8 to cause apoptosis (caspases are proteolytic enzymes causing death) [43]. An essential characteristic of NASH is the over-expression of these death receptors and the resulting apoptosis [44].

### 2.3. Mitochondrial Dysfunction and Reactive Oxygen Species

Cell damage due to reactive oxygen species (ROS), a class of free radicals generated from molecular oxygen, is called "oxidative stress" [45]. Although the mitochondria are a major source of ROS and are generated by oxidative reactions within cells, levels of ROS are very low in healthy cells due to a variety of antioxidant defensive mechanisms [46,47]. The liver prefers to eliminate FFAs through mitochondrial β-oxidation in healthy individuals [48]. However, in NAFLD, there is an excess of FFAs, and the increase in mitochondrial β-oxidation translates into an increase in electron supply at the electron transport chain; this causes the electron transport chain to be over reduced and ROS to develop [49]. Given that mitochondrial DNA is likely to be damaged by ROS, a higher production of ROS results in mitochondrial malfunction, increasing the risk of ROS formation [50]. The release of proapoptotic proteins such as cytochrome c into the cytosol is produced by mitochondrial dysfunction caused by intracellular stress caused by an accumulation of ROS [51]. Apoptosome is an active complex formed when cytochrome c, apoptotic-protein activation factor-1 (Apaf-1), and caspase 9 attach [52]. Caspases 3, 6, and 7 downstream are activated by apoptosome to achieve the remaining stages of apoptosis [53].

### 2.4. Lysosomal Permeabilization

The investigation of molecular mechanisms makes it possible to discover the lysosomal−mitochondrial axis in FFA-induced lipotoxicity and the potential role of lysosomic permeability in the progression of NASH [54]. Mitochondrial dysfunction is considered to be the principal pathophysiologic process contributing to the progression of NALFD into NASH [55]. In human liver cells, lysosomal permeabilization and the release of the lysosomal protease cathepsin B occurred far earlier than mitochondrial dysfunction and cytochrome c release into the cytosol [56]. In addition, the inhibition of cathepsine B protects against lipotoxicity caused by FFA [4]. By activating hepatic stellate cells and

promoting their development in the myofibroblasts of mice, cathepsin B is also related to the evolution of hepatic fibrosis [57].

*2.5. Endoplasmic Reticulum Stress*

An intracellular organelle called the endoplasmic reticulum (ER) carries out many critical tasks such as protein and lipid production [58]. When the ER is stressed (ER stress), it responds with a process known as an unfolded protein response (UPR) [59]. UPR aims to protect ER from stress caused by a number of factors, including viral infections, alcohol, and FFAs [60]. However, if stress in the ER goes on for a long time, UPR might be unable to handle it, causing apoptosis [61]. In vitro research, using different models of exocrine pancreas cells, has shown how saturated fatty acid, such as palmitic acid, could produce ER stress and hepatic cell death, furthering our understanding of the role of ER stress [62]. Apoptosis can also to be caused by FFAs in other ways, including mitochondrial dysfunction caused by the activation of c-Jun N-terminal kinase (JNK), mitochondrial permeabilization caused by the pro-apoptotic protein BAX, free cholesterol-mediated ER stress, and ceramide-mediated apoptosis induced by death ligands such as TNF and FAS [63] (Figure 1).

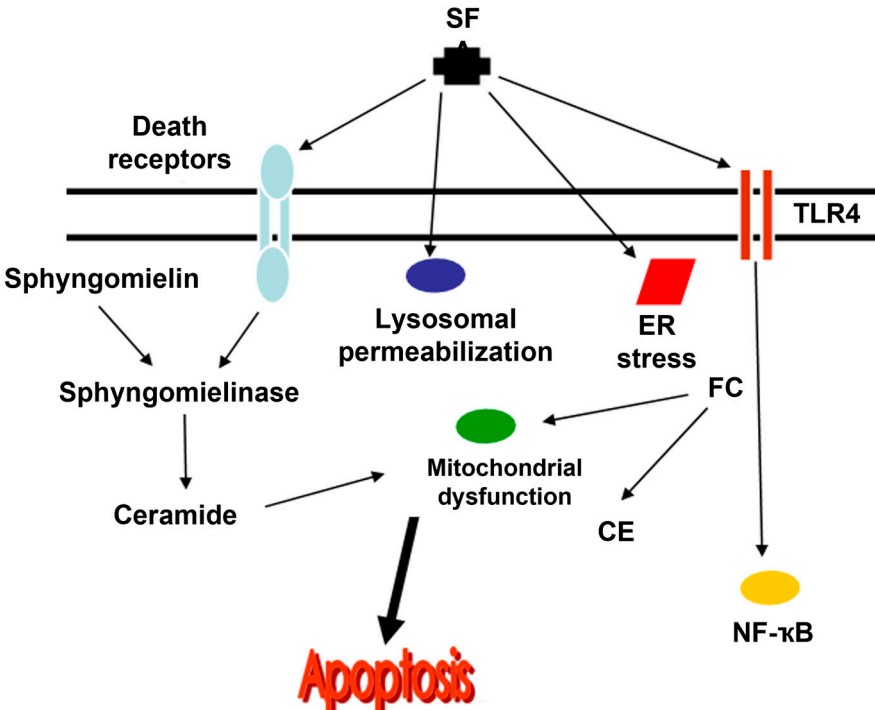

**Figure 1.** Possible mechanisms through which FFAs can lead to apoptosis. Lysosomal permeabilization, ER stress, and mitochondrial malfunction are all associated with the activation of the mitochondrial route of apoptosis. FFA may also upregulate and increase the amount of death receptors in the plasma membrane, such as Fas and TRAIL receptor 5 (DR5). These harmful fatty acids may also stimulate TLR4 signaling, which will increase the production of a number of cytokines that promote inflammation. Moreover, other lipid forms including ceramide and free cholesterol (FC) may cause mitochondrial malfunction and trigger the apoptotic pathway in the mitochondria. Abbreviations: SFA: saturated fatty acids; FC: free cholesterol; CE: cholesteryl-ester; ER: endoplasmic reticulum.

## 3. Pleiotropic Signals of Hepatotoxicity and Lipids

Lipids are a structurally diverse class of hydrophobic molecules that carry out a variety of functions. Moreover, lipids are essential energy-storing molecules that help form cellular membranes, are involved in a number of signal transduction cascades, and have a caloric yield of 9 kcal/g compared with 4 kcal/g for proteins and carbohydrates [64]. Ceramides, fatty acids, leukotrienes, and prostaglandins are a few of the lipid groups that have been

identified to mediate hepatotoxic effects while causing hepatic inflammation [65]. The liver plays an important role in the systemic metabolism of lipids.

### 3.1. Prostaglandins Cyclooxygenase 2

Prostaglandins, produced by (COX2), are associated with a number of physiological and pathological processes, including cell growth and the induction of inflammatory responses [66]. The majority of non-steroidal anti-inflammatory drugs (NSAIDs), including well-known NSAIDs such as acetylsalicylic acid, ibuprofen, and sulindac, perform de facto as reasonably all-purpose COX inhibitors from the production of prostaglandins from arachidonic acid. The majority of non-steroidal anti-inflammatory drugs (NSAIDs), including well-known NSAIDs such as acetylsalicylic acid, ibuprofen, and sulindac, act de facto as reasonable COX inhibitors, inhibiting the synthesis of prostaglandins from arachidonic acid [67]. The role of COX2 in hepatic disorders is still debatable at the moment [68]. On the one hand, it has been shown that COX2 hepatocyte-specific transgenic expression accelerates D'galactosamine/LPS-induced liver failure by a prostaglandin E (PGE) receptor 1, subtype EP1 (PTGER1)-dependent mechanism involving the activation of JNK2 [69], and PGE2 has been associated with NAFLD and NASH through causative mechanisms [70]. However, there is strong genetic evidence that COX2 protects mice against lethal hepatitis caused by agonistic anti-FAS antibodies, probably due to its control of epidermal growth factor (EGFR) receptor expression [71]. In a manner similar to this, prostaglandins, in particular PGE2, appear to have potent hepatoprotective effects against IR and acetaminophen hepatotoxicity [72].

### 3.2. Leukotrienes

The principal final products of arachidonic acid metabolism are prostaglandins and leukotrienes, which are produced by the catalytic activity of the enzyme arachidonate 5-lipoxygenase (ALOX5) [73]. Leukotrienes participate in inflammatory reactions, cause chemotaxis, regulate the contraction of tracheal smooth muscles, and contribute to the development of asthma [74]. According to reports, the deletion of Alox5 or treatment with CCl4 reduces spontaneously emerging steatotic, inflammatory, and fibrotic responses in Apoe-/- mice [75,76]. In several distinct NASH and NAFLD models, it has been found that pharmacological ALOX5 inhibitors or FLAP, an accessory protein of ALOX5, can reduce the liver damage caused by CCl4 and prevent the inflammatory infiltration of the hepatic parenchyma [77], possibly because they stimulated Kupffer cells to die off preferentially. Arachidonate 15-lipoxygenase, which functions similarly to ALOX5 in the metabolism of arachidonic acid, may have a key role in the emergence of NASH and NAFLD [78]. In fact, the arachidonate changes and increased expression of enzymes involved in eicosanoid and prostanoid production are consistent with the development and severity of inflammation during NAFLD/NASH development.

#### 3.2.1. Ceramides

Ceramides are lipid messengers that accumulate because of a variety of stressors. They can do this either through a three-step synthesis pathway, which mainly occurs in ER membranes, or by sphingomyelinases hydrolyzing sphingomyelin [79]. Ceramides are now understood to be important regulators of systemic metabolism, because of their well-known ability to regulate the amounts of proliferation and apoptosis that are specific to each cell [80,81]. Ceramides accumulate and enhance direct mitochondrial effects [82], promote the formation of ROS [83], and inhibit a number of anti-apoptotic signal transducers, including AKT [84], particularly when the death receptor is activated [82,83]. Although mice lacking acidic sphingomyelinase (aSMase) are significantly protected from these effects [85], and it has been shown that the pharmacological aSMase inhibitor imipramine limits liver damage after IR injury [86]; D-galactosamine/LPS, TNFa, and agonistic anti-FAS antibodies all exhibit lethal hepatotoxic effects. In a similar way, myriocin, an inhibitor of de novo ceramide synthesis, appears to have advantages in genetically obese (ob/ob) mice

and animals fed a high-fat diet, including a decrease in body weight, an improvement in metabolism, and an increase in energy expenditure [87].

Notably, myriocin has also been demonstrated to inhibit the replication of the hepatitis C virus (HCV) in vivo [88], suggesting that ceramides may function as the primary regulators of hepatotoxicity in a variety of clinical situations.

### 3.2.2. Fatty Acids and Neutral Fats

Fatty acids and neutral fats, such as mono-, di-, and tri-glycerides, serve as key energy substrates and essential components of cell membranes, but they also function as precursors for the synthesis of a variety of other substances [89]. However, numerous tissues, such as the liver, are toxic when fatty acids and neutral fats accumulate as a result of metabolic problems (such as insulin resistance, which is typically accompanied by an increase in circulating fatty acids) or an inadequate diet [90]. In these conditions, fatty acids and neutral fats (mostly di- and tri-glycerides) accumulate within hepatocytes, as well as in response to different toxins and xenobiotics such as ethanol, causing a distinctive change in the hepatic parenchyma, known as steatosis [91]. Steatosis is associated with (at least to some extent) liver damage as a result of inflammation and the pro-apoptotic activity of free fatty acids (FFAs), regardless of its underlying etiology [92]. Observations from a few studies using obese mice with leptin deficiency (ob/ob) and leptin receptor deficiency (db/db), as well as many in vitro lipotoxicity models, suggest that the hepatotoxic effects of FFAs are not mediated by RIPK1-dependent signal pathways, but rather by mitochondrial apoptosis and death-receptor-transduced signals [93,94]. In particular, TRAIL receptor overexpression [95]; ROS production [96]; BAX binding with p-MLKL mediating lysosomal membrane permeabilization [97]; activation of the tumor suppressor protein p53 [98]; and direct mitochondriotoxic effects induced by the local accumulation of modified lipids, such as oxidized cardiolipin [99] and ceramides [100], appear to be involved in FFA-induced apoptosis. It has been shown, in support of this hypothesis, that a combination of saturated fatty acids, such as palmitic and oleic acid, increases the sensibility of hepatocytes to the induction of cell death by a variety of stimuli [101,102]. FFAs increased the expression of PTEN, BAX, and miR-181a-5p in Huh7 cells while decreasing the expression of Akt phosphorylation, XIAP, and Bcl2 [103]. When miR-181a-5p pre-miRs were transfected into Huh7 cells, similar effects were observed; however, these alterations were reversed in FFA-treated, anti-miR-181a-5p-transfected Huh7 cells [103]. While PTEN and BAX expression was increased in the CDAA-fed mice, Akt phosphorylation, XIAP, and Bcl2 were all significantly decreased [103]. In mice administered CDAA, the miR-181a-5p expression was also considerably higher [103]. Therefore, these results suggest that free fatty acids elevated miR-181a-5p in hepatic cells, which caused to their apoptosis [103].

In the human body, triclosan (TCS) and triclocarban (TCC) have been found at concentrations as high as hundreds of nanomolars [104]. Previous studies focused on the pollutant itself, which had little to no effect on the hepatic lipid buildup caused by TCS and TCC [104]. A high-fat diet (HFD), a known environmental factor contributing to diseases involving lipid metabolism, merits consideration due to its synergistic impact along with environmental pollutants [104].

In light of this, it would be too simplified to assume that all fats are harmful to the liver. Despite the fact that the lipotoxic potential of FFAs as a class of lipids is well known, (a) specific FFAs vary in their ability to cause cell death, with saturated and long lipids typically being considered to be more toxic than their non-saturated and short counterparts [105], and (b) neutral fats, in particular triglycerides, seeming to be rather passive, if not triggering an adaptive response with hepatoprotective effects [106].

### Lipid Receptors

*Low-density lipoprotein receptor (LDLR).*

Low-density lipoprotein receptor (LDLR) is an integral membrane protein that is most abundantly expressed in the liver, where it binds to and removes LDL-C from circulation

by endocytosis [107]. Therefore, the amount of LDLR expressed in hepatocytes is inversely correlated with the level of plasma LDL-C. LDL receptors are present in the cell membranes of liver cells (hepatocytes) and other cells throughout the body [108]. They make cholesterol accessible to healthy bodily cells. LDLs release their cholesterol and triglycerides after binding to LDL receptors on hepatocytes. Although practically, all tissues express LDLR, the liver is a key organ for absorbing plasma LDL-c as it removes over 70% of it [109]. The level of plasma LDL-c is reduced by increased hepatic LDLR expression, offering a method for treating hypercholesterolemia. Familial hypercholesterolemia is a type of elevated cholesterol that is brought on by mutations in the LDLR gene. In this gene, more than 2000 mutations have been found. Some of these genetic alterations cause cells to create fewer low-density lipoprotein receptors [110].

*The LDL receptor-related protein-1 (LRP1).*

The apoE receptor, which functions in conjunction with the LDL receptor in the liver to eliminate chylomicron and VLDL residual lipoproteins from the blood, was initially identified as the LDL-receptor-related protein-1 (LRP1) [111]. Since its initial identification, LRP1 has also been shown to be highly expressed in a variety of cell types, interact with a broad range of macromolecular substrates, and control cell functions in a cell type- and context-dependent manner. Several studies revealing that LRP1 gene polymorphisms are connected with a broad spectrum of disorders serve as the finest illustration of the significance of these LRP1 roles in health maintenance and disease protection [112,113]. However, we do not yet fully understand the underlying mechanisms through which LRP1 malfunction encourages obesity, diabetes, and fatty liver disease.

Studies using conditional knockout mice with tissue-specific LRP1 loss or in vitro cell culture experiments provide the majority of knowledge about how LRP1 failure may increase the risk of metabolic diseases [114]. Studies on the function of LRP1, which is expressed in hepatocytes, have shown that it is critical for chylomicron and VLDL remnant absorption, HDL secretion, and apoA-I lipidation [111,115].

The expression of LRP1 in hepatocytes is also required to limit fat-induced hepatic steatosis, lipotoxicity, insulin resistance, and steatohepatitis [116,117]. In adipose tissues, LRP1 has been shown to be essential for the maturation of preadipocytes into mature adipocytes, as well as for the appropriate storage of lipids by these mature adipocytes [118,119]. Diet-induced obesity and insulin resistance are decreased by the inactivation of LRP1 in mature white and brown adipocytes by shifting lipid resources to the muscle to enhance thermogenesis [120]; however, hyperlipidemic animals with normal LRP1 expression in the artery wall also have increased inflammation and accelerated atherosclerosis due to the lack of LRP1 in mature adipocytes [121].

*VLDL receptor (VLDLR).*

Members of the LDL receptor (LDLR) gene family and the multifunctional VLDL receptor (VLDLR) have structural similarities [122]. VLDLR comprises five distinct protein domains, namely, a single transmembrane domain, an O-linked glycosylation sugar domain, an extracellular N-terminal ligand-binding region with eight cysteine-rich repeats, and a cytoplasmic domain containing the NPxY motif for signal transduction [123]. The low-density lipoprotein receptor (LDLR) has five cysteine-rich repeats, whereas the human VLDLR has only one [123]. These two variants of the receptor are revealed by the isolation and characterization of the cDNAs encoding human VLDLR [123]. Additionally, VLDLR alternative splicing produces a variety of transcript variants that each encode a different isoform; however, the expression of these proteins and their precise activities have not yet been determined [123]. In addition to macrophages, VLDLR is widely expressed in the heart, skeletal muscle, adipose tissue, endothelium, and brain [124,125]. While not naturally present in the liver, some circumstances such as endoplasmic reticulum stress and treatment with the PPARa agonist fenofibrate can promote hepatic expression. The expression of VLDLR is insulin-dependent [126] and is unaffected by cellular cholesterol levels [127], in contrast with LDLR.

### 4. Effects of Lipid-Altering Drugs on Hepatotoxicity

Symptoms of drug-induced hepatotoxicity caused by lipid-altering drugs include acute liver failure, hepatitis, cholestasis, and increased liver enzymes (hypertransaminasemia) [128] (Table 1). Data indicate that small asymptomatic increases in aspartate transaminase (AST) and alanine transaminase (ALT) do not always predict acute liver failure; in fact, increased AST and ALT levels are not specific or exclusive to liver damage [129]. Possible liver damage is indicated by unusual weakness or fatigue, appetite loss, upper abdominal pain, dark urine, and a yellowing of the skin or eye whites (jaundice) [130]. Statins, bile acid resins (BARs), fibrates, analogues of fibric acid, inhibitors of cholesterol absorption, niacin, and fish oil are the main categories of lipid-altering drugs [131]. There are very few cases of liver damage with the use of fish oil and BARs. Red yeast rice (RYR) is another dietary supplement that could possibly change lipid levels [132]. The few cases of hepatotoxicity associated with this medication will also not be mentioned. Physicians should take appropriate action when using RYR because it often contains various amounts of lovastatin [133]. There are numerous pathways through which cholesterol compounds can have hepatotoxic effects. Statins, for example, are metabolized in the liver after being absorbed in the gastrointestinal (GI) tract, in contrast with BARs and the cholesterol absorption inhibitor ezetimibe, which primarily target the GI tract, but also have a little effect on the liver [134].

**Table 1.** Overview of lipid-altering agents.

| Drug(s) | Key Contraindication | Recommended Liver Function Monitoring |
|---------|----------------------|---------------------------------------|
| Statins | Active or chronic liver disease | Obtain AST/ALT initially, 12 weeks after starting, then annually or sooner if clinically indicated. Baseline, with follow-up, only as clinically indicated |
| Fibrates | Gall bladder disease, hepatic disease (biliary cirrhosis), or severe renal impairment including dialysis | Liver tests should be monitored periodically |
| Ezetimibe | Active liver disease or unexplained AST/ALT elevations (when co-administered with a statin) | When co-administered with statin therapy, monitor according to recommendations for individual statins |
| Niacin | Active liver disease, active peptic ulcer disease, or severe gout | Obtain AST/ALT initially, 6–8 weeks after reaching 1500 mg daily, 6–8 weeks after reaching max daily dose, then annually or sooner if clinically indicated |

#### 4.1. Statins

Statins (including atorvastatin, fluvastatin, lovastatin, pitavastatin, pravastatin, rosuvastatin, and simvastatin) provide observable reductions in LDL-C by inhibiting 3-hydroxy-3-methylglutaryl-coenzyme A (HMG-CoA) reductase, the rate-limiting step in cholesterol synthesis [135]. This regulation reduces the creation of cholesterol, which increases the expression of LDL receptors and accelerates the clearance of LDL-C from circulation. Statins are still the first-line treatment, despite their modest impact on triglyceride reduction and HDL-C elevation. This regulation decreases the creation of cholesterol, which increases the expression of LDL receptors and removes LDL-C from circulation rapidly. Statins are still the first-line treatment despite their limited impact on triglyceride reduction and HDL-C elevation. This is because they lower LDL-C. Statins significantly reduce major coronary events and overall mortality among populations receiving treatment for primary and secondary prevention, according to significant findings from a number of clinical trials [136,137]. In addition, statins improve endothelial function, decreasing inflammatory markers and stabilizing atherosclerotic plaque [136,137].

The results of the clinical trials show a minimal overall risk of hepatotoxicity during statins use. The most frequent hepatic adverse events are asymptomatic increases in ALT and AST. Generally, during the first year of taking treatment, this dose-dependent response may appear at any time [138]. Studies have shown that alterations are often reversible with a reduced dose and can be returned to normal using the same dosage [139].

The Adult Treatment Panel III (ATP-III) guidelines of the National Cholesterol Education Program (NCEP) include detailed recommendations for assessing hepatic function, specifically transaminase elevation [140]. Patients with AST/ALT values below three times the upper limit of normal (ULN) may continue to be treated with statins [141]. If elevations are more than three times the ULN, a second evaluation of the liver function should be carried out [142]. If the high quantities of AST/ALT persist, the statin must be stopped, while switching to another statin or rechallenge may also be possible [143]. Transaminase levels that were overall high and higher than the ULN were present in only 2% of patients receiving statin monotherapy [144].

Isolated case reports have described statin-related cirrhosis, autoimmune hepatitis, fulminant hepatitis, and cholestatic hepatotoxicity [145,146]. The National Lipid Association's (NLA) Liver Expert Panel suggests obtaining a fractionated bilirubin level if the transaminase levels are high and statin-induced hepatotoxicity is expected [147]. When there is no biliary blockage, bilirubin is a more accurate indicator of drug-induced liver injury [148]. More testing needs to be completed to determine the reason for this, as elevated transaminase and bilirubin levels probably indicate ongoing liver injury [142]. The risk of hepatotoxicity in older patients can be increased by taking high dosages of statins, CYP450 enzyme inhibitors, or inducers concurrently, or combining lipid-altering regimens, resulting in impaired renal function [149].

The value of systematic monitoring of transaminase in treated patients has been questioned because statins rarely result in long-lasting liver injury [141]. Regular monitoring raises concerns about additional costs and the possibility that statin therapy may be inappropriately interrupted due to elevated levels of transaminase [141]. The perception that patients have possible liver injury leads to statin rejection and non-compliance, which is worrisome [141].

The FDA changed the statin prescription by removing the requirement to routinely test liver enzymes in accordance with the NLA Liver Expert Panel's recommendations [150]. The FDA decided that continuous monitoring contributed little to no function for identifying or treating a major liver injury; therefore, baseline testing is still advised, but follow-up monitoring is only required if clinically warranted [151].

*4.2. Fibrates*

Fibrates (such as clofibrate, fenofibrate, gemfibrozil, and fenofibric acid) are frequently administered to decrease triglycerides; typical reductions vary from 25% to 50% [152]. The agents have varying effects on LDL-C [153]. Fibrates frequently increase LDL-C in hypertriglyceridemia, although there may be slight improvements in normal triglyceride levels [153]. Although fibrates and statins are routinely combined, there have been cases of elevated transaminase levels and increased risk of myopathy [154]. Gemfibrozil increases the serum levels of the majority of statins [155]. Furthermore, if the concurrent statin dose is kept low to moderate, adverse side effects, such as hepatotoxicity, often remain at a low [141]. Within weeks of stopping the medicine, transaminase levels normalized in cases of transaminase elevations [151]. Gemfibrozil increases the serum levels of the majority of statins [155]. In addition, if the concomitant dose of statins is maintained at a low to moderate level, adverse side effects, such as hepatotoxicity, often remain low [141]. A few weeks after discontinuing the drug, transaminase levels are normalized in cases of transaminase elevation [151].

The NLA or NCEP ATP-III guidelines do not mention liver function testing in connection with fibrate treatment [152]. Therefore, it is important to take baseline measurements, monitor monthly, and reduce or stop taking drugs when the transaminase levels are greater than three times the ULN [142]. The risk of hepatotoxicity can also be increased by a number of conditions, such as drug interactions and pre-existing liver diseases, in addition to other considerations that physicians should take into account.

*4.3. Ezetimibe*

Ezetimibe reduces LDL-C by inhibiting cholesterol absorption at the gut's brush border [156]. This drug is not processed by the CYP450 enzyme system and does not interact with CYP3A4 inhibitors. Ezetimibe is glucuronidated after absorption to produce an active metabolite that circulates through enterohepatic recirculation [157].

Because of its rarity, there are no specific guidelines available for monitoring hepatic function when taking ezetimibe monotherapy [158]. Ezetimibe−simvastatin combination therapy may cause in slightly higher transaminase levels than simvastatin alone [159]. There is no appreciable difference in transaminase concentrations between the safety outcomes of atorvastatin−ezetimibe and atorvastatin monotherapy [160]. Similar to statin therapy, it is crucial to monitor the liver function of those who take ezetimibe together with a statin, and to take measurements at baseline and then as needed, for example during dose titration [161].

*4.4. Niacin*

Niacin, also known as vitamin B3 or nicotinic acid, improves all major lipid biomarkers (including LDL-C, non-HDL, and triglycerides), when given at the right therapeutic doses [162]. Niacin is also the most effective medication for increasing HDL-C [162]. With niacin therapy, patients with CHD whose LDL-C is within the target range may show regression in the mean carotid intima-media thickness [161]. However, findings from clinical studies indicate that niacin supplementation did not further reduce the frequency of vascular events when combined with statin medications [162].

There are three primary types of niacin, namely immediate release (IR) or crystalline, extended release (ER), and sustained release (SR), with corresponding absorption rates of 1, 8, and 12 h, respectively [162]. There is a great deal of misunderstanding about the different dosages; medication administration methods; effects on the lipid profile; and potential side effects, including hepatotoxicity. Liver function should be checked on all niacin formulations.

The side effects of IR niacin include flushing, chills, pruritus, GI distress, and a very low risk of hepatotoxicity [162]. Extended-release niacin is associated with decreased flushing rates and a low incidence of hepatotoxicity at doses 2000 mg/day [162]. However, SR is frequently accompanied by higher transaminase levels that are dose-dependent [162].

The differences in hepatotoxicity between formulations are likely caused by two distinct metabolic pathways [160]. The combination of nicotinic acid, which has a high capacity but a low affinity, with glycine results in flushing. Niacin is transformed into nicotinamide using a second nonconjugative pathway, which is a high-affinity, low-capacity process with a higher risk of hepatotoxicity. The majority of the drug will be metabolized through conjugation, resulting in greater flushing and a low incidence of hepatotoxicity, as the nonconjugative pathway will be quickly saturated with IR products [162]. The predominant metabolism of slowly absorbed formulations, such as SR niacin, by the high-affinity nonconjugative pathway, results in modest flushing and increased hepatotoxicity.

## 5. Discussion

Insulin resistance, with the result that the liver stores fat, especially triacylglycerol, characterizes NAFLD. The steatotic liver exhibits hepatocyte apoptosis, oxidative stress, and low-grade liver damage. However, liver damage, hepatocyte death, elevated oxidative stress, and liver inflammation progress more quickly in some patients. The mechanisms that induce steatosis to evolve to steatohepatitis are probably diverse and complex, comparable to other environmental diseases such as alcoholic steatohepatitis or lung cancer in smokers.

We advocate the following hepatotoxicity model for fatty livers: In hepatocytes susceptible to steatosis, circulating FFA can trigger a variety of intracellular responses (Figure 2). These include JNK activation, TLR4 activation, BAX activation, lysosomal permeabilization, and ER stress. If these changes are large enough, hepatocyte apoptosis and mitochondrial permeabilization are probable outcomes. Steatohepatitis patients with NAFLD have consid-

erably higher rates of hepatocyte apoptosis than patients with simple steatosis. The genesis of progressive steatohepatitis may be caused by increased vulnerability to apoptosis in fatty hepatocytes, such as via the regulation of the Bcl-2 gene family or through changes in JNK activity. However, people who develop progressive steatohepatitis may have selective activation of the ER stress pathway. The development of diagnostic and therapeutic strategies aimed at reducing the morbidity and mortality rates associated with NAFLD should be encouraged by developments in the comprehension of the molecular pathways behind liver damage.

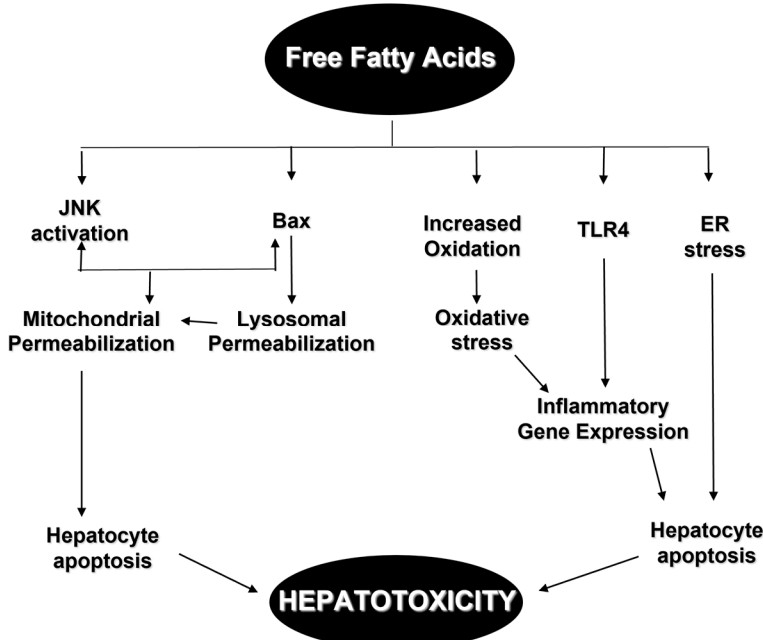

**Figure 2.** Free fatty acids (FFAs) and hepatotoxicity. FFA toxicity manifests itself at several molecular levels. C-jun N-terminal kinase (JNK) can be activated by free fatty acids, and the degree of activation is correlated with their toxicity. BAX activates JNK, which causes the mitochondria to become more permeable. Apoptosis and lysosomal permeabilization are two additional outcomes of BAX. Oxidative stress and antioxidant depletion are caused by increased oxidation of fatty acids, thus facilitating the development of reactive oxygen species. Palmitic acid activates toll-like receptor 4 (TLR 4), which controls the expression of inflammatory genes. In humans, NAFLD/NASH also exhibits endoplasmic reticulum stress, which can be brought on by saturated fatty acids.

Many mechanisms cause liver cell destruction and exacerbate existing damaged processes. The mitochondria are frequently the target of several drugs that have hepatotoxic effects. The dysfunction of these important cellular organelles, which also leads to intracellular oxidative stress with an increase in the generation of reactive oxygen species and peroxynitrite, impairs energy metabolism. In addition to the mitochondria, cytochrome P450 isoenzymes encourage oxidative stress and cellular damage. The accumulation of biliary acids causes additional stress and cytotoxicity when hepatocellular function is impaired. Additionally, Kupffer cells are activated and neutrophils are drawn into the liver from cellular damage, a gut-derived endotoxin, or a combination of both. Hepatotoxicity is still an important factor in the removal of drugs from clinical usage and in pharmaceutical development due to the numerous direct and indirect pathways that produce drug-induced cell injury in the liver.

## 6. Conclusions

As previously mentioned, hepatic steatosis, which is defined by an excessive accumulation of TG in hepatocytes, is affected by lipogenesis and fatty acid oxidation. On the other hand, the development of steatohepatitis, activation of stellate cells, and progressive and continuous deposition of hepatic fibrosis leading to cirrhosis include the death of liver cells and the regeneration of surviving hepatocytes. However, it is important to understand the molecular pathways underlying the hepatotoxicity by an excess of lipids in the liver and NAFLD. Metabolic syndrome, which is clearly associated with a diet high in saturated fats and refined sugars (such as fructose and/or glucose), is characterized by insulin resistance and glucose intolerance. The disparity of the energy storage system is disturbed by an excess of FFA and carbohydrates, which results in cytotoxicity to the tissues.

In most lipid-altering drugs, severe drug-induced hepatotoxicity is extremely rare. Furthermore, the incidence rises when specific drugs and additional risk factors are present. Hepatotoxic episodes can be minimized with appropriate surveillance. It will be necessary to ask the correct questions and gather the required information to determine the risk for hepatotoxicity associated with lipid-altering medications. Identifying the conditions that may increase a patient's risk for liver damage and compiling a complete list of all prescription medications and dietary supplements to check for potential interactions are crucial pieces of information.

The prevalence of liver disease is constantly increasing in industrialized countries due to a number of lifestyle variables, including excessive caloric intake, unbalanced diet, lack of physical activity, and abuse of hepatotoxic medicines. Considering the important functions of cell death and inflammation in the etiology of the majority, if not all, liver diseases, one efficient therapeutic treatment may include the administration of hepatoprotective and anti-inflammatory drugs, either alone or in combination.

Clinical trials are currently being conducted in cohorts of patients with different liver diseases to explore this theory.

**Funding:** This research received no external funding.

**Institutional Review Board Statement:** Not applicable.

**Informed Consent Statement:** Not applicable.

**Conflicts of Interest:** The author declares no conflict of interest.

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
