# Peer review of "Molecular Mechanisms and Mediators of Hepatotoxicity Resulting from an Excess of Lipids and Non-Alcoholic Fatty Liver Disease"

_gastrointestdisord, doi:10.3390/gidisord5020020_

Round 1

Reviewer 1 Report

The review by Finelli describes role of excess lipids in hepatotoxicity in the context of NAFLD. The review article is well-written and will be highly referred in future.

Minor comments:

1)      Figure 1 is illegible. A high resolution (300 dpi) figure should be included.

2)      Although Author indicate hepatotoxicity model for fatty liver in Figure 2 – it is missing from the manuscript.

3)      Minor grammatic errors should be fixed.

4)      A comprehensive table indicating various effect of regulation of lipids in NAFLD/NASH  will be beneficial to the readers.

Author Response

I thank the reviewers for their time to review our manuscript and providing valuable comments. I have considered each of the comments made and done the necessary changes to the best of our effort and understanding. A detailed response to each of the reviewers’ comments is presented below.

Reviewer 1

The review by Finelli describes role of excess lipids in hepatotoxicity in the context of NAFLD. The review article is well-written and will be highly referred in future.

Minor comments:

  • Figure 1 is illegible. A high resolution (300 dpi) figure should be included.

Author reply: This has been corrected.

  • Although Author indicate hepatotoxicity model for fatty liver in Figure 2 – it is missing from the manuscript.

Author reply: Figure 2 has been enclosed

  • Minor grammatic errors should be fixed.

Author reply: This has been corrected.

  • A comprehensive table indicating various effect of regulation of lipids in NAFLD/NASH  will be beneficial to the readers.

Author reply: See Figure 2

Reviewer 2 Report

Dott. Finelli presents a review on molecular mechanisms.

While I appreciate the amount of dedicated work put into this review, I think that it needs significant restructuring. 

- Many sentences / thought are mentioned directly twice, indicating that the text has not been carefully re-read after the final draft was written.

- the manuscript spans a huge variety of topics which are connected loosely. One part may be the molecular mechanisms of lipid excess, another the therapy of hyperlipidemia, another the (not very relevant hepatotoxicity) of lipid-lowering drugs, which however are not used to treat hepatic liver overload in the first place. The author should either focus or connect these dots better.

 - the details of the manuscript have not always been worked out. E.g., the figure  does not mention most of the molecular pathways indicated in the text referring to the figure. Some sentences are rather unprecise: "Arachidonate.... may have a key role in the emergence of NASH and NAFLD": why? what key role? In NASH or rather NAFLD?

Author Response

I thank the reviewers for their time to review our manuscript and providing valuable comments. I have considered each of the comments made and done the necessary changes to the best of our effort and understanding. A detailed response to each of the reviewers’ comments is presented below.

Reviewer 2:

Dott. Finelli presents a review on molecular mechanisms.

While I appreciate the amount of dedicated work put into this review, I think that it needs significant restructuring. 

- Many sentences / thought are mentioned directly twice, indicating that the text has not been carefully re-read after the final draft was written.

Author reply: This has been corrected

- the manuscript spans a huge variety of topics which are connected loosely. One part may be the molecular mechanisms of lipid excess, another the therapy of hyperlipidemia, another the (not very relevant hepatotoxicity) of lipid-lowering drugs, which however are not used to treat hepatic liver overload in the first place. The author should either focus or connect these dots better.

Author reply: This has been corrected.

 - the details of the manuscript have not always been worked out. E.g., the figure  does not mention most of the molecular pathways indicated in the text referring to the figure. Some sentences are rather unprecise: "Arachidonate.... may have a key role in the emergence of NASH and NAFLD": why? what key role? In NASH or rather NAFLD?

Author reply: This has been specified

Reviewer 3 Report

The review article entitled "Molecular Mechanisms and Mediators of Hepatotoxicity by an Excess of Lipids in the Liver and Non-Alcoholic Fatty Liver Disease" comprehends the insights of liver damage by various factors.

The review article needs substantial revisions.

1. the title needs to be revised as "hepatotoxicity" explains liver damage so no need to include liver again in the title.

2. abstract is exactly the same as written in the introduction. The abstract should demonstrate the comprehensiveness of the review article. 

3. Introduction needs to revised extensively. Introduction to be started with NAFLD.

4. The third paragraph from TLR "A fundamental component of NASH is the lipotoxicity of hepatocytes. Lipotoxicity is caused by the accumulation of lipid intermediates which cause..........." to be moved to introduction fro clarity.

5. The author does not talk anything about the lipid receptors such as LDLR,LRP-1, VLDLR, SR1B.....

6. a schematic events for each mediators separately should be shown for better understanding.

7. Table of drugs, their mode of action with relevant references to be placed in the article.

Author Response

I thank the reviewers for their time to review our manuscript and providing valuable comments. I have considered each of the comments made and done the necessary changes to the best of our effort and understanding. A detailed response to each of the reviewers’ comments is presented below.

Reviewer 3:

The review article entitled "Molecular Mechanisms and Mediators of Hepatotoxicity by an Excess of Lipids in the Liver and Non-Alcoholic Fatty Liver Disease" comprehends the insights of liver damage by various factors.

The review article needs substantial revisions.

  1. the title needs to be revised as "hepatotoxicity" explains liver damage so no need to include liver again in the title.

Author reply: This has been corrected

  1. abstract is exactly the same as written in the introduction. The abstract should demonstrate the comprehensiveness of the review article. 

Author reply: This has been corrected

  1. Introduction needs to revised extensively. Introduction to be started with NAFLD.

Author reply: This has been corrected.

  1. The third paragraph from TLR "A fundamental component of NASH is the lipotoxicity of hepatocytes. Lipotoxicity is caused by the accumulation of lipid intermediates which cause..........." to be moved to introduction fro clarity.

Author reply: This has been corrected.

  1. The author does not talk anything about the lipid receptors such as LDLR, LRP-1, VLDLR, SR1B.....

Author reply: This has been specified

  1. a schematic events for each mediators separately should be shown for better understanding.

Author reply: See Figure 2

  1. Table of drugs, their mode of action with relevant references to be placed in the article.

Author reply: The table has been introduced

Round 2

Reviewer 2 Report

None

Author Response

I thank the reviewer for his time to review our manuscript and providing valuable comments. 

Reviewer 3 Report

The author has mede extensive changes. The article looks to be in good shape than the first version. However i find the abstract needs some improvement?

The authors have used the same lines represented in the introduction first lines to appear on the abstract.

The abstract needs to be improvised reflecting the comprehensiveness of the article. 

Author Response

I thank the reviewer for his time to review our manuscript and providing valuable comments. I have considered the comments made and done the necessary changes to the best of our effort and understanding.

Abstract: The paper reviews some of the mechanisms implicated in the hepatotoxicity induced by excess of lipids. The paper spans a huge variety of topics: one part is the molecular mechanisms of lipid excess, another the therapy of hyperlipidemia, another hepatotoxicity of lipid-lowering drugs. NAFLD is now the leading cause of chronic liver disease in western countries, the molecular mechanisms leading to NAFLD are only partially understood, and there are not effective therapeutic interventions. The prevalence of liver disease is constantly increasing in industrialized countries due to a number of lifestyle variables, including excessive caloric intake, an unbalanced diet, a lack of physical activity and the abuse of hepatotoxic medicines. Considering the important functions of cell death and inflammation in the etiology of the majority, if not all, liver diseases, one efficient therapeutic treatment may include the administration of hepatoprotective and anti-inflammatory drugs, either alone or in combination. Clinical trials are currently being conducted in cohorts of patients with different liver diseases to explore this theory.